# Comparing Humans, GPT-4, and GPT-4V On Abstraction and Reasoning Tasks

**Melanie Mitchell, Alessandro B. Palmarini, and Arseny Moskvichev**

Santa Fe Institute
1399 Hyde Park Road
Santa Fe, NM 87501
mm@santafe.edu, abp@santafe.edu, arseny.moskvichev@gmail.com

## Abstract

We explore the abstract reasoning abilities of text-only and multimodal versions of GPT-4, using the ConceptARC benchmark (Moskvichev, Odouard, and Mitchell 2023), which is designed to evaluate robust understanding and reasoning with core-knowledge concepts. We extend the work of Moskvichev et al. by evaluating GPT-4 on more detailed, one-shot prompting (rather than simple, zero-shot prompts) with text versions of ConceptARC tasks, and by evaluating GPT-4V, the multimodal version of GPT-4, on zero- and one-shot prompts using image versions of the simplest tasks. Our experimental results support the conclusion that neither version of GPT-4 has developed robust abstraction abilities at humanlike levels.

## Introduction

To what extent have large pre-trained language models (LLMs) developed "emergent" capabilities for abstract reasoning? The defining characteristic of abstract reasoning is the ability to induce a rule or pattern from limited data or experience and to apply this rule or pattern to new, unseen situations. Such abilities are a key aspect of human intelligence; even very young children are adept at learning abstract rules from just a few examples (Walker and Gopnik 2014).

Recently, various researchers have claimed that sufficiently large pre-trained language models can develop emergent abilities for reasoning (Wei et al. 2022), general abstract pattern recognition (Mirchandani et al. 2023), and analogy-making (Webb, Holyoak, and Lu 2023). However, the internal mechanisms giving rise to these abilities are not well understood, and other researchers have cast doubt on the claims that these systems actually form humanlike abstractions (Gendron et al. 2023), showing in many cases that while LLMs can solve problems involving content similar to that in their training data, they are weak in generalizing outside such problems (McCoy et al. 2023; Razeghi et al. 2022; Wu et al. 2023). Some have interpreted this as evidence that LLMs rely not on generalizable abstract reasoning but on learning complex patterns of associations in their training data and performing "approximate retrieval" of these patterns in new situations (Kambhampati 2023).

Abilities for creating and reasoning with abstract representations are fundamental to robust generalization, so it is essential to understand the extent to which LLMs have achieved such abilities. In this paper we report on experiments evaluating GPT-4 on tasks in ConceptARC (Moskvichev, Odouard, and Mitchell 2023), a collection of analogy puzzles that test general abstract reasoning capabilities. We show that by providing prompts with more detailed instructions and a simple solved example, GPT-4's performance on a text version of this corpus improves substantially above that reported in previous work, but remains substantially below that of humans and of special-purpose algorithms for solving tasks in this domain. Because humans are given these tasks in a visual modality, it has been argued that it would only be fair to compare humans with multimodal (rather than text-only) LLMs. We perform this comparison using GPT-4V, the multimodal extension of the GPT-4, and show that this particular multimodal LLM performs substantially worse than the text-only version. These results reinforce the conclusion that a large gap in basic abstract reasoning still remains between humans and state-of-the-art AI systems.

## The Abstraction and Reasoning Corpus

Chollet (2019) proposed the Abstraction and Reasoning Corpus (ARC) as a benchmark for fairly evaluating such abilities in both humans and machines. ARC consists of 1,000 manually created analogy puzzles ('tasks"), each of which contains a small number (typically 2–4) demonstrations of transformations on grids, and a "test input" grid. The task for the solver is to induce the abstract rule underlying the demonstrations and to apply that rule to the test input to generate a transformed grid. Figure 1 gives three examples of ARC tasks.

According to Chollet, the prior knowledge needed for solving these tasks is a subset of the core knowledge systems hypothesized to be innate in humans (Spelke and Kinzler 2007)—namely, *objectness, numerosity, and basic geometry and topology*. Notably, Chollet intentionally omitted knowledge of language or other "learned symbols" from the required prior knowledge, which helps avoid any "approximate retrieval" and pattern matching based on prior training data that might underlie LLMs' apparent success on language-based reasoning tasks. Instead, ARC is meant

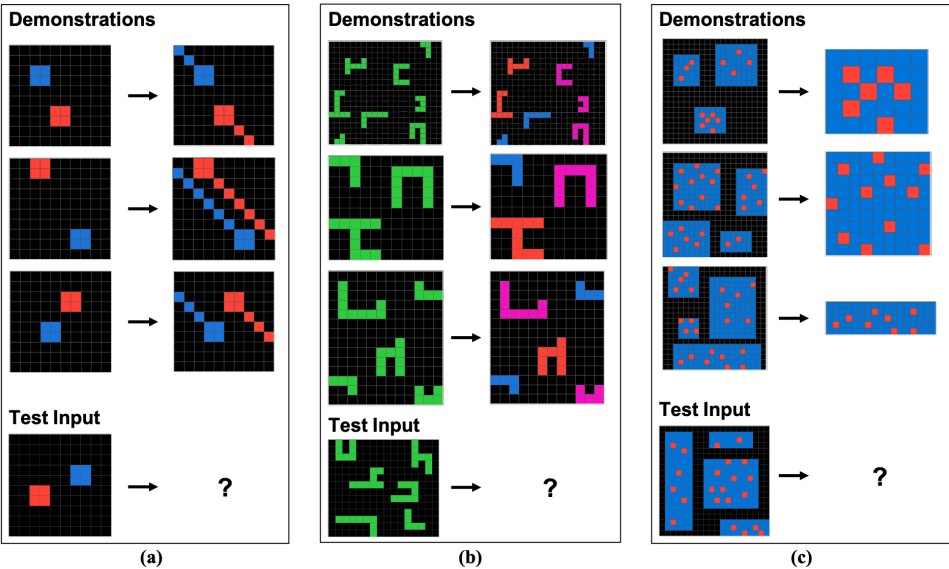

Figure 1: Examples of ARC tasks from Chollet (2023). Each task has a set of demonstration input-output pairs that illustrate an abstract grid-transformation rule, and a test input. The solver's challenge is to generate a new grid that results from applying the abstract rule to the test input. (Figure is from Moskvichev et al. (2023); best viewed in color.)

to capture the crux of abstract reasoning: inducing general rules or patterns from small numbers of examples and applying these flexibly to new, previously unseen situations.

Chollet (2023) published 800 ARC tasks and kept the remaining 200 as a "hidden" test set. 100 of these hidden tasks were used as an test set for a challenge on the Kaggle platform (Kaggle.com 2020). The first-place program from the Kaggle challenge solved 21% of these hidden tasks, and an ensemble of the first- and second-place programs solved 31%. This remains the highest-achieved accuracy on ARC to date. The winning programs on Kaggle used program-synthesis methods that searched over combinations of manually defined, primitive grid operations in order to find combinations that correctly map inputs to outputs in the task demonstrations. The authors of the winning programs acknowledge that such methods are not likely to generalize well (Wind 2020; de Miquel Bleier 2020) and ARC remains largely unsolved by any AI methods.

Several groups have tested LLMs on subsets of ARC tasks (Gendron et al. 2023; Mirchandani et al. 2023; Xu et al. 2023; Wang et al. 2023), using different prompting formats, and generally found the best accuracy using straightforward text versions of tasks to be around 10–12%. Limited studies of human performance on subsets of ARC tasks has shown much higher accuracies (e.g., 84% in Johnson et al. 2021).

## Previous Evaluation of LLMs using the ConceptARC Benchmark

Moskvichev et al. (2023) noted two problems with the original ARC corpus. First, they claimed, many of the tasks are quite difficult, even for humans, and this difficulty might be a barrier to progress in developing AI systems that reason in this domain. Second, and most important, ARC does not of-

fer *systematic* evaluation of understanding of particular core concepts; even if a system can solve an individual ARC task, that does not necessarily mean that the system has a robust understanding of the underlying concepts. To address these issues, Moskvichev et al. created a new benchmark in the ARC domain, ConceptARC, whose tasks are intentionally designed to be easy for humans and, moreover, whose 480 tasks are organized as systematic variations of particular core spatial and semantic concepts, such as *Top and Bottom, Inside and Outside, Center, and Same and Different*. Each concept group contains 30 tasks, each of which instantiates the concept in a different way, and with differing degrees of abstraction. Moskvichev et al.'s claim was that high performance over these various instantiations of a given concept indicates a robust understanding of, and ability to reason abstractly about, the underlying concept.

The authors gave these tasks to human participants on the Amazon Mechanical Turk and Prolific platforms. They also tested the two winning programs from the Kaggle ARC challenge as well as GPT-4 on all 480 tasks. They found that human performance substantially exceeded that of machines on all concept groups in the corpus; in particular, the overall accuracy of humans was 91%, compared to the first-place Kaggle program's accuracy of 52%, and GPT-4's accuracy of 25% (using temperature 0.5). The authors concluded that "[t]he generally high accuracies of humans on each concept indicates successful generalization over the different variations in each given concept group. In contrast, the much lower accuracies of programs we tested indicates a lack of ability to generalize over the variations in a concept group, and thus a failure to develop the abstractions that ARC is meant to test."

Moskvichev et al.'s evalation of GPT-4 had two important

limitations: (1) the prompt format that they used was overly simple and might not have communicated enough about the task; and (2) it might not be fair to compare the performance of humans, who are presented with a visual version of each task, with LLMs, which are given a text-only version of the task.

Here we address both of these limitations, through two sets of experiments. In the first set of experiments, we evaluate the text-only version of GPT-4 on text versions of ConceptARC tasks using a much more expressive prompt that includes both instructions and an example of a solved task, making this a one-shot rather than zero-shot induction problem. In the second set of experiments, we evaluate GPT-4V, the multimodal version of GPT-4, on the visual version of the simplest ConceptARC tasks, giving it a similar prompt as in the first set of experiments but using images rather than text to represent tasks.

## Experiments Evaluating Text-Only GPT-4 on ConceptARC Tasks

To evaluate the text-only version of GPT-4, we adapted the prompt used in a recent study by Wang et al. (2023). The exact prompt is given in the Supplementary Information. This prompt provides detailed instructions about the task as well as an example of a solved task. If GPT-4 responds with an incorrect answer, we repeat a request for it to supply a different answer, for a maximum of three guesses, which is standard for ARC evaluations (Kaggle.com 2020). If a correct answer is generated within these three guesses, the task is considered to be solved.

We used this prompting method with OpenAI's API to test GPT-4[1] on all 480 ConceptARC tasks (30 per each of the 16 concept groups)[2], first with GPT-4's temperature set to zero and then with temperature set to 0.5, to test the effects of temperature on performance. The accuracies (fraction of solved tasks within each concept group as well as fraction of solved tasks overall) are given in Table 1, along with the human accuracies from the Moskvichev et al. (2023) study (details of the human studies are given in that paper). Note that, like GPT-4, humans are given three guesses for each task.

For GPT-4, our more detailed, one-shot prompting method resulted in a higher accuracy overall, 0.33 for both temperature settings, than the simple zero-shot method used by Moskvichev et al. (2023), which reported 0.19 for temperature zero and 0.25 for temperature 0.5. However, GPT-4's performance remains well below the high performance of humans, supporting the conclusion of Moskvichev, Odouard, and Mitchell (2023) that, even with more informative prompting, the system lacks basic abstract reasoning abilities tested by this corpus.

## Experiments Evaluating GPT-4V on Minimal ConceptARC Tasks

We did a second set of experiments to test the hypothesis that GPT-4V, the multimodal version of GPT-4, would obtain higher performance than the text-only version. Due to the costs of running such an experiment on the 480 tasks in ConceptARC, we decided to establish a baseline for comparing text-only GPT-4 and GPT-4V using the *minimal tasks* created by Moskvichev et al. (2023). For each of their 16 concept groups, Moskvichev et al. created three minimal tasks—extremely simple instantiations of the concept[3]. They used these tasks as "attention checks" in their human studies, to make sure that human participants were paying attention and understood the basic idea of the tasks. Participants who failed at solving two or more minimal tasks (out of three given) were excluded from the study. Moskvichev et al. reported that 12 out of 482 participants were excluded on this basis.

Though it was not reported in their paper, we obtained the data from Moskvichev et al. on human performance on minimal tasks from all 482 participants in their study. We included the previously excluded 12 participants because, unlike Moskvichev et al., we are evaluating human performance on the minimal problems, so even if two or more of these problems were "failed" by participants, we need to include that in our evaluation here. We also ran the text-only version of GPT-4 on these tasks, using the same prompting method described in the previous section. As shown in Table 2, the fraction of humans correctly solving the 48 minimal tasks was 0.95, and GPT-4's accuracies were 0.69 (temperature 0) and 0.65 (temperature 0.5). The difference between GPT-4's performance on these minimal problems and on the non-minimal tasks (Table 1) underline the simplicity of the minimal tasks compared to those in the regular corpus.[4]

We explored various approaches to presenting the minimal tasks to GPT-4V[5]: displaying all input-output pairs of a single task within one image, using a separate image for each input-output pair, and providing each input and each output grid as an individual image. Only the last approach yielded correct solutions during our preliminary investigations, and is thus the approach we adopted. Furthermore, when presented with an image, GPT-4V was unable to consistently translate the visual grid to a text representation, including both color names and numeric encodings. Therefore, to mitigate errors involved in mapping the intended output grid to a text representation, we requested only a natural language description of the grid, along with its dimensions.

We aimed to quantify the influence of visual representa-

---

[1]We used the version of GPT-4 (gpt-4-0613) available in November, 2023.

[2]The tasks can be downloaded from https://github.com/victorvikram/ConceptARC

[3]The minimal tasks can be downloaded from https://github.com/victorvikram/ConceptARC

[4]For comparison, the accuracy of the first-place Kaggle-ARC program was 0.81 on the minimal tasks versus 0.52 on the regular corpus.

[5]We used the version of GPT-4V available in November, 2023 (gpt4-vision-preview). This version does not allow a manual temperature setting and the documentation does not specify the default temperature.

| Concept | Humans | GPT-4 $Temp = 0$ | GPT-4 $Temp = 0.5$ |
|---|---|---|---|
| Above and Below | 0.90 | 0.50 | 0.47 |
| Center | 0.94 | 0.37 | 0.37 |
| Clean Up | 0.97 | 0.43 | 0.46 |
| Complete Shape | 0.85 | 0.47 | 0.40 |
| Copy | 0.94 | 0.37 | 0.33 |
| Count | 0.88 | 0.27 | 0.23 |
| Extend To Boundary | 0.93 | 0.20 | 0.20 |
| Extract Objects | 0.86 | 0.13 | 0.13 |
| Filled and Not Filled | 0.96 | 0.27 | 0.30 |
| Horizontal and Vertical | 0.91 | 0.33 | 0.37 |
| Inside and Outside | 0.91 | 0.30 | 0.33 |
| Move To Boundary | 0.91 | 0.23 | 0.17 |
| Order | 0.83 | 0.27 | 0.30 |
| Same and Different | 0.88 | 0.23 | 0.30 |
| Top and Bottom 2D | 0.95 | 0.60 | 0.63 |
| Top and Bottom 3D | 0.93 | 0.30 | 0.27 |
| **All concepts** | **0.91** | **0.33** | **0.33** |

Table 1: Accuracies of humans and GPT-4 (with temperature 0 and 0.5) on each concept group (30 tasks) and over all concepts (480 tasks) in ConceptARC. The results on humans are from Moskvichev, Odouard, and Mitchell (2023).

| **Humans** | **GPT-4** $Temp = 0$ | **GPT-4** $Temp = 0.5$ | **GPT-4V Zero-Shot** | **GPT-4V One-Shot** |
|---|---|---|---|---|
| 0.95 | 0.69 | 0.65 | 0.25 | 0.23 |

Table 2: Accuracies of humans, GPT-4 (with Temperature 0 and 0.5), and GPT-4V (zero- and one-shot prompting) on minimal tasks over all concepts (48 tasks) in ConceptARC.

tions on performance by maintaining consistency with our text-only evaluation. The same prompting method and example of a solved task was used, modified only by substituting text representations of grids with visual counterparts. This prompt, along with the variations from the text-only approach, are given in the Supplementary Information.

GPT-4V often included descriptions of an abstract transformation rule as part of its solution. Our assessment focused exclusively on the accuracy of the model's test output grid descriptions. In certain cases, the model accurately described the output grid despite identifying an incorrect abstract rule, which we classified as a success. On the other hand, we classified as failures instances in which the model correctly identified the abstract rule but failed to accurately describe the output grid. Note that these inconsistencies support the arguments of West et al. (2023) that LLMs' generative abilities can be disconnected from the ability to "understand" the language they generate.

The results of our experiments with GPT-4V on minimal tasks are given in the last two columns of Table 2. The visual one-shot prompt resulted in an accuracy of 0.23 over the 48 minimal tasks, compared with 0.69 for the (temperature 0) text-only counterpart. Notably, several of GPT-4V's unsuccessful output grid descriptions incorporated details from the solved example, suggesting that including an sample solved task in the prompt may have had a negative impact on performance. Consequently, we extended our evaluation of GPT-4V to include a zero-shot setting.

The zero-shot prompt is also given in the Supplemen-

tary Information. Unlike the one-shot setting, the evaluation was done using OpenAI's web application. This required a slightly different prompting method, where the model was able to respond with observations after receiving each demonstration input-output pair, and before receiving the test input grid to provide its solution. The zero-shot prompting method resulted in an accuracy of 0.25 on minimal tasks, solving one additional task compared to the one-shot setting, and with an overlap of seven tasks successfully solved in both settings.

While we tested GPT-4V only on minimal tasks, We expect that GPT-4V's overall performance would be similarly considerably worse than the text-only version on the much-harder non-minimal-task corpus.

## Conclusion

In this paper we extended work done by Moskvichev et al. (2023) on evaluating the abstract reasoning capabilities of GPT-4, using the ConceptARC corpus, which systematically tests abstraction abilities using basic core concepts. Moskvichev et al. found that GPT-4 had substantially worse performance than both humans and the first-place program in the Kaggle-ARC challenge on these tasks. However, the prompting method they used was overly simple, and they experimented only with text versions of the tasks. Here, we performed evaluations using a more informative, one-shot prompt for text versions of tasks, and experimented with similar zero- and one-shot prompts for the multimodal case in which task-grids were given as images. We found that our

more informative one-shot prompt improved GPT-4's performance in the text case, but its performance remained well below that of humans and the special-purpose Kaggle-ARC program. We also found that giving minimal tasks as images to the multimodal GPT-4 resulted in substantially worse performance than in the text-only case. Our results support the hypothesis that GPT-4, perhaps the most capable "general" LLM currently available, is still not able to robustly form abstractions and reason about basic core concepts in contexts not previously seen in its training data. It is be possible that other methods of prompting or task representation would increase the performance of GPT-4 and GPT-4V; this is a topic for future research.

**Acknowledgments** This material is based in part upon work supported by the National Science Foundation under Grant No. 2139983. Any opinions, findings, and conclusions or recommendations expressed in this material are those of the authors and do not necessarily reflect the views of the National Science Foundation. This work has also been supported by the Templeton World Charity Foundation, Inc. (funder DOI 501100011730) under the grant https://doi.org/10.54224/20650.

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

# Supplementary Information

## Prompts for Text-only GPT-4

In their evaluation of GPT-4, Moskvichev et al. (2023) translated ConceptARC tasks into text representations used in prompts like the one shown in Figure 2. Here the 10 possible colors are encoded as integers, and each row of a grid is encoded as a list of integers inside square brackets.

Figure 3 shows an example of the prompt we used (adapted from Wang et al. (2023) in testing text-only GPT-4. We use the format required by the OpenAI API. The symbol "#" indicates comments not given in the actual prompt. We used the same encoding for colors and grid rows as Moskvichev et al.

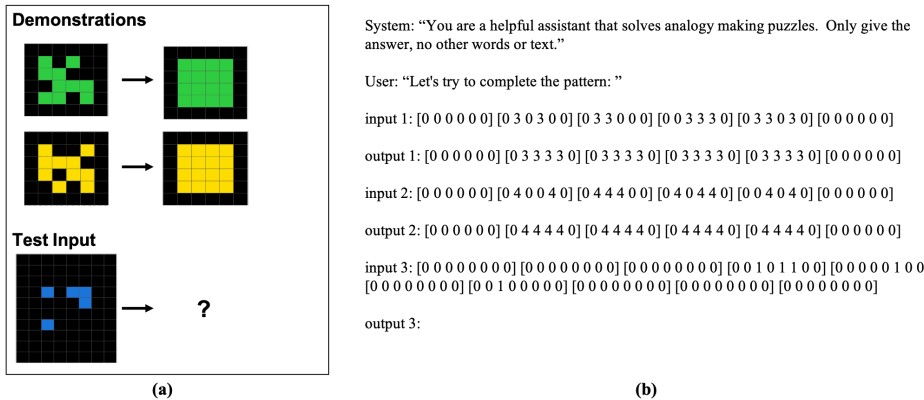

**(a)**

System: "You are a helpful assistant that solves analogy making puzzles. Only give the answer, no other words or text."

User: "Let's try to complete the pattern: "

input 1: [0 0 0 0 0 0] [0 3 0 3 0 0] [0 3 3 0 0 0] [0 0 3 3 3 0] [0 3 3 0 3 0] [0 0 0 0 0 0]

output 1: [0 0 0 0 0 0] [0 3 3 3 3 0] [0 3 3 3 3 0] [0 3 3 3 3 0] [0 3 3 3 3 0] [0 0 0 0 0 0]

input 2: [0 0 0 0 0 0] [0 4 0 0 4 0] [0 4 4 4 0 0] [0 4 0 4 4 0] [0 0 4 0 4 0] [0 0 0 0 0 0]

output 2: [0 0 0 0 0 0] [0 4 4 4 4 0] [0 4 4 4 4 0] [0 4 4 4 4 0] [0 4 4 4 4 0] [0 0 0 0 0 0]

input 3: [0 0 0 0 0 0 0 0] [0 0 0 0 0 0 0 0] [0 0 0 0 0 0 0 0] [0 0 1 0 1 1 0 0] [0 0 0 0 0 1 0 0] [0 0 0 0 0 0 0 0] [0 0 1 0 0 0 0 0] [0 0 0 0 0 0 0 0] [0 0 0 0 0 0 0 0] [0 0 0 0 0 0 0 0]

output 3:

**(b)**

Figure 2: (a) A task from the ConceptARC corpus. (b) The corresponding prompt used by Moskvichev et al. (2023) to give to GPT-4. (From Moskvichev et al. (2023); best viewed in color.)

## Prompts for GPT-4V

Figure 4 shows the adapted prompt used to test GPT-4V in the one shot setting, for the same example used in Figure 3. Differences from the text-only prompt are highlighted in red text.

Figure 5 shows an example of the prompts used to test GPT-4V in the zero-shot setting. This evaluation was conducted using OpenAI's web application and thus all messages are sent as the 'user' role, , interspersed with the model providing a response after each message.

```
# GENERAL INSTRUCTIONS
[System]
You will be given a list of input-output pairs, labeled "Case 0", "Case 1", and so on. Each input and output is a grid of
numbers representing a visual grid. There is a SINGLE rule that transforms each input grid to the corresponding output grid.

The pattern may involve counting or sorting objects (e.g. sorting by size), comparing numbers (e.g. which shape or symbol
appears the most? Which is the largest object? Which objects are the same size?), or repeating a pattern for a fixed number
 of time.

There are other concepts that may be relevant.
- Lines, rectangular shapes
- Symmetries rotations, translations.
- Shape upscaling or downscaling, elastic distortions.
- Containing / being contained / being inside or outside of a perimeter.
- Drawing lines, connecting points, orthogonal projections.
- Copying, repeating objects.

You should treat cells with 0 as empty cells (backgrounds).

Please generate the Output grid that corresponds to the last given Input grid, using the transformation rule you induced
from the previous input-output pairs.

# EXAMPLE SOLVED TASK
[User]
Case 0:
Input:
[0 0 0 0 0 0] [0 2 2 0 0 0] [0 2 2 0 0 0] [0 0 0 0 0 0] [0 0 0 0 0 0] [0 0 0 0 0 0]
Output:
[0 0 0 0 0 0] [0 0 2 2 0 0] [0 0 2 2 0 0] [0 0 0 0 0 0] [0 0 0 0 0 0] [0 0 0 0 0 0]

Case 1:
Input:
[0 0 0 0 0 0] [0 0 4 4 0 0] [0 0 4 4 0 0] [0 0 0 0 0 0] [0 0 0 0 0 0] [0 0 0 0 0 0]
Output:
[0 0 0 0 0 0] [0 0 0 4 4 0] [0 0 0 4 4 0] [0 0 0 0 0 0] [0 0 0 0 0 0] [0 0 0 0 0 0]

Case 2:
Input:
[0 0 0 0 0 0] [0 3 0 0 0 0] [0 3 3 3 0 0] [0 0 0 3 0 0] [0 0 0 0 0 0] [0 0 0 0 0 0]
Output:
[0 0 0 0 0 0] [0 0 3 0 0 0] [0 0 3 3 3 0] [0 0 0 0 3 0] [0 0 0 0 0 0] [0 0 0 0 0 0]

Case 3:
Input:
[0 0 0 0 0 0] [0 0 0 0 0 0] [0 0 6 0 0 0] [0 0 6 0 0 0] [0 0 6 6 0 0] [0 0 0 0 0 0]

[Assistant]
Output:
[0 0 0 0 0 0] [0 0 0 0 0 0] [0 0 0 6 0 0] [0 0 0 6 0 0] [0 0 0 6 6 0] [0 0 0 0 0 0]

# QUERY TASK
[User]
Case 0:
Input:
[0 0 0 0 0 0] [0 3 0 3 0 0] [0 3 3 0 0 0] [0 0 3 3 3 0] [0 3 3 0 3 0] [0 0 0 0 0 0]
Output:
[0 0 0 0 0 0] [0 3 3 3 3 0] [0 3 3 3 3 0] [0 3 3 3 3 0] [0 3 3 3 3 0] [0 0 0 0 0 0]

Case 1:
Input:
[0 0 0 0 0 0] [0 4 0 0 4 0] [0 4 4 4 0 0] [0 4 0 4 4 0] [0 0 4 0 4 0] [0 0 0 0 0 0]
Output:
[0 0 0 0 0 0] [0 4 4 4 4 0] [0 4 4 4 4 0] [0 4 4 4 4 0] [0 4 4 4 4 0] [0 0 0 0 0 0]

Case 2:
Input:
[0 0 0 0 0 0] [0 6 0 0 6 6] [0 6 0 6 0 0] [0 6 0 6 0 0] [0 0 6 0 6 6] [0 0 6 6 6 0]

[Assistant]
Output:

# IF THE SYSTEM RETURNS WRONG ANSWER (REPEAT UP TO TWO TIMES)
[User]
Your answer does not solve the puzzle. Try again.

[Assistant]
I apologize for my mistake. Here is a better answer:
Output:
```

Figure 3: Example of the prompt used to test text-only GPT-4 on ConceptARC tasks. The symbol "#" indicates comments not given in the actual prompt.

```
# GENERAL INSTRUCTIONS
[System]
You will be given a list of input-output pairs, labeled "Case 0", "Case 1", and so on. Each input and output is a 2D visual
 grid containing colored cells. There is a SINGLE rule that transforms each input grid to the corresponding output grid.

The pattern may involve counting or sorting objects (e.g. sorting by size), comparing numbers (e.g. which shape or symbol
appears the most? Which is the largest object? Which objects are the same size?), or repeating a pattern for a fixed number
 of time.

There are other concepts that may be relevant.
- Lines, rectangular shapes
- Symmetries rotations, translations.
- Shape upscaling or downscaling, elastic distortions.
- Containing / being contained / being inside or outside of a perimeter.
- Drawing lines, connecting points, orthogonal projections.
- Copying, repeating objects.

You should treat black cells as empty cells (backgrounds).

Please provide a language description of the Output grid that corresponds to the last given Input grid, using the
transformation rule you induced from the previous input-output pairs. Please include the dimensions of the Output grid as
part of your description.

# EXAMPLE SOLVED TASK
[User]
Case 0:
Input:
[Image of input grid]
Output:
[Image of output grid]

Case 1:
Input:
[Image of input grid]
Output:
[Image of output grid]

Case 2:
Input:
[Image of input grid]
Output:
[Image of output grid]

Case 3:
Input:
[Image of input grid]

[Assistant]
Output:
The output should contain the same pink object found in the input, shifted one cell to the right. The dimensions of the
output grid should be 6x6.

# QUERY TASK
[User]
Case 0:
Input:
[Image of input grid]
Output:
[Image of output grid]

Case 1:
Input:
[Image of input grid]
Output:
[Image of output grid]

Case 2:
Input:
[Image of input grid]

[Assistant]
Output:

# IF THE SYSTEM RETURNS WRONG ANSWER (REPEAT UP TO TWO TIMES)
[User]
Your answer does not solve the puzzle. Try again.

[Assistant]
I apologize for my mistake. Here is a better answer:
Output:
```

Figure 4: Example of the prompt used to test GPT-4V in the one-shot setting. The symbol "#" indicates comments not given in the actual prompt. Red text signifies differences from the prompt used to test text-only GPT-4, as provided in Figure 3.

```
# GENERAL INSTRUCTIONS
[User]
You will be given a sequence of images containing several cases of input-output pairs. Each input and output is a 2D visual
 grid containing colored cells. There is a SINGLE pattern that transforms each input grid to the corresponding output grid.

The pattern may involve counting or sorting objects (e.g. sorting by size), comparing numbers (e.g. which shape or symbol
appears the most? Which is the largest object? Which objects are the same size?), or repeating a pattern for a fixed number
 of time.

There are other concepts that may be relevant.
- Lines, rectangular shapes
- Symmetries rotations, translations.
- Shape upscaling or downscaling, elastic distortions.
- Containing / being contained / being inside or outside of a perimeter.
- Drawing lines, connecting points, orthogonal projections.
- Copying, repeating objects.

You should treat black cells as empty cells (backgrounds).

I will present each example input-output pair one at a time. The input will always be the first attached image and the
output will always be the second attached image. You can note down any observations about the possible transformation after
 seeing each input-output example. Once I have shown you all the input-output examples, I will present you a test input
grid and ask you to provide a language description the single pattern that transformed each example input and a language
description of the test output grid when using the pattern on the test input grid. Please include the dimensions of the
output grid as part of your description.

# QUERY
[User]
[Image of input grid] [Image of output grid]

[User]
[Image of input grid] [Image of output grid]

[User]
[Image of input grid]
Here is the test input grid. Provide a language description the single pattern that transformed each example input and a
language description of the test output grid when using the pattern on the test input grid. Please include the dimensions
of the output grid as part of your description.

# IF THE SYSTEM RETURNS WRONG ANSWER (REPEAT UP TO TWO TIMES)
[User]
Incorrect. Try again.
```

Figure 5: Example of the prompts used to test GPT-4V in the zero-shot setting. The symbol "#" indicates comments not given in the actual prompt.