# OpenReview forum: "Comparing Humans, GPT-4, and GPT-4V On Abstraction and Reasoning Tasks"
_AAAI.org/2024/Workshop/LLM-CP — LLM-CP @ AAAI 2024 Oral_

### Official Review · Reviewer_hjQm · 2023-12-04
**Evaluating humans, GPT-4, and GPT-4V on ConceptARC**

**Rating:** 2
**Confidence:** 2

**Review:**

The authors evaluate GPT-4 text only, GPT-4V, and humans on the ConceptARC benchmark which consists of various grid analogy puzzles testing abstract reasoning. It's exciting to see a systematic evaluation of GPT-4V on a reasoning task. I think the results make for interesting discussion and will add to the workshop. However, I have a few comments/questions:
- How about evaluating humans on the text-only version?
- Would appreciate more discussion on the comparison between zero-shot and one-shot on GPT4V. “Notably, several of GPT-4V’s unsuccessful output grid descriptions incorporated details from the solved example, suggesting that including an sample solved task in the prompt may have had a negative impact on performance.” — this leakage is interesting and should be analyzed further. Why was zero-shot done on the web interface but one-shot done with the API?
- Have the authors tried a grid reference system like columns are 1-6, rows are A-F, describe the grid with statements like “cell D3 is green”, etc.?
- ConceptARC seems to focus specifically on spatial arrangement type abstract reasoning. Could the authors say more about the scope of the work? How might GPT4(V) compare on other types of non-spatial abstract reasoning?

---

### Official Review · Reviewer_FYvu · 2023-12-07
**Significant work but**

**Rating:** 2
**Confidence:** 2

**Review:**

The paper evaluates the performance of the GPT-4 and GPT-4V (multimodal version of GPT-4) on abstraction and reasining tasks from Concept ARC. Their evaluations on the minimal tasks as well as evaluations using elaborated one-shot prompts indicates that these large models are still very far from human level concept abstraction and understanding. The paper provides empirical evidence to what many researchers have claimed and speculated, and hence is a significant contribution.


Pros:
* The article is well written, demonstrates clarity and provides all the required details.
* The evaluations are thorough and significant.

Cons:
* The contribution of the paper is not directly relevant to the theme of the workshop. However, the ability of abstraction and concept understanding is important for causal reasoning, hence this might still be relevant for the workshop audience.

Questions/Comments
* It is not clear why the 12 participants that were excluded in the Moskvichev et al. were included in this evaluation.
* It is interesting that  GPT-4V performs better with zero-shot prompt than one-shot prompt. How significant is that performance difference?

---

### Official Review · Reviewer_UjtY · 2023-12-07
**The authors apply OpenAI's GPT-4 and GPT-4V on the ConceptARC benchmark: a grid-based few-shot learning benchmark designed to test abstract reasoning**

**Rating:** 2
**Confidence:** 2

**Review:**

This paper has everything that I want to see in a workshop submission:

- results on a relevant topic (how does the recent GPT4 / GPT4-V perform on the ConceptARC benchmark
- good problem motivation, relevant background, two tables of results, and an appendix to answer some of my follow-up questions

The goals are clear, the experiments are (fairly) easy to understand, and the interpretation of the results is fair.

---

## Feedback and Room for Improvement

### 1: Reporting GPT-4V Results

The biggest room for improvement lies in how the GPT-4V results get reported. Near the end of page 4 the authors describe:

> "GPT-4V often included descriptions of the abstract transformation rule as part of its solution ..."

This seems interesting in itself. But the authors deal with this observation with:

> "In certain cases, the model accurately described the output grid despite identifying an incorrect abstract rule, which we classified as a success. On the other hand, we classified as failures instances in which the model correctly identified the abstract rule but failed to accurately describe the output grid."

This observation--that GPT-4V can achieve a correct *final answer* despite generating tokens that are sometimes correct or incorrect--appears closer to what the authors set out to answer: how do humans ... and GPT-4V compare on abstract reasoning tasks?

Does ConceptARC define a set of "ground truth" labels for what each abstract rule is? If so: this could be an extra experiment set to improve on the paper's core goal. If ConceptARC does not: a strong "+1" contribution may be how effectively human participants and LLM-style models can describe the abstract rule.

### 2: Prompt Gradation for Core Concepts

The top of page 2 motivated the ConceptARC tasks as having:

> "... systematic variations of particular core and semantic concepts, such as *Top and Bottom*, *Inside and Outside*, *Center*, and *Same and Different*.

However, the prompts shown in the appendix do not appear to highlight all of these in the "System" settings. There is a point about how the concept of "*Containing / being contained / being inside or outside of a perimeter*" may be a relevant concept (which maps onto the "Inside and Outside" concept); others like "Top and Bottom" are less clear.

I see two possible routes:

1. Describe how the concepts were translated into the system parameters
2. Further delineate experiments by "increasingly strong hints in the system parameters". Such as: (1) providing little or no system parameters hints, (2) providing general hints, (3) providing text descriptions of the core concepts, (4) providing a description of the abstract rule.

Route (2) may be more helpful toward understanding the failure cases. I suspect there would be a stronger trend between *increasing background information* and *output accuracy*, which may be a stronger motivation toward answering the cases where LLMs fail to reach an answer even when the prompter provides most relevant information.

---

### Official Review · Reviewer_HVCs · 2023-12-07
**The paper investigates GPT-4's abstract reasoning capabilities using the ConceptARC benchmark, comparing text-only and multimodal versions. It builds upon prior research by assessing GPT-4's performance on more intricate, one-shot prompts within ConceptARC tasks using text input. Additionally, it evaluates GPT-4V on zero- and one-shot prompts using image-based tasks. They found that neither GPT-4 version has achieved robust abstraction abilities at human-like levels.**

**Rating:** 2
**Confidence:** 2

**Review:**

The paper builds upon previous research by addressing key limitations in Moskvichev et al.'s evaluation of GPT-4. Firstly, it improves the evaluation by introducing a more detailed and expressive prompt format for the text-only version of GPT-4.  But prompt is a very subjective method. How can we evaluate whether the prompts used in the paper are specific enough to bring out the ability of GPT-4?

---

### Meta-Review · Area_Chair_k5LK · 2023-12-10

**Recommendation:** 2
**Confidence:** 3

**Metareview:**

All reviewers are very positive about the paper, seeing it as an interesting and clear contribution, thus I think the paper would be a great addition to the workshop. There is room though for further clarification, and the authors are also encouraged to take inspiration from the reviewers' suggestions on further experiments, so I hope the authors take the reviews to heart.

---

### Decision · Program_Chairs · 2023-12-14

**Decision:**

Accept (Oral)

**Comment:**

Thank you for submitting your work to the LLM-CP workshop @ AAAI 2024. See you in Vancouver!